# The Impact of an Onboarding Plan for Newly Hired Nurses and Nursing Assistants: Results of a Quasi-Experimental Study

**DOI:** 10.3390/nursrep15110398

**Published:** 2025-11-12

**Authors:** Pilar Montes Muñoz, Pablo Cardinal-Fernández, Ángel Morales Rodríguez, Cayetana Ruiz-Zaldibar, Alicia de la Cuerda López

**Affiliations:** 1Department of Nursing, Centro Universitario HM Hospitales de Ciencias de la Salud (CUHMED), Universidad Camilo José Cela, 28660 Madrid, Spain; pcardinal@hmhospitales.com (P.C.-F.); amrodriguez@hmhospitales.com (Á.M.R.); crzaldibar@ucjc.edu (C.R.-Z.); adelacuerda@hmhospitales.com (A.d.l.C.L.); 2Instituto de Investigación Sanitaria HM Hospitales, 28015 Madrid, Spain; 3Department of Nursing, Faculty of Medicine, Universidad CEU San Pablo, CEU Universities, 28003 Madrid, Spain; 4Intensive Care Unit, HM Torrelodones University Hospital, 28250 Madrid, Spain

**Keywords:** onboarding process, nursing management, safety, nursing satisfaction

## Abstract

**Background/Objectives:** High turnover and staff shortages in nursing pose challenges to professional integration and compromised patient safety. Structured onboarding programs are considered key strategies to enhance adaptation, reduce clinical errors, and promote retention. This study aimed to evaluate the impact of a structured onboarding program compared with the standard routine on early professional adaptation, safety culture, and satisfaction among newly hired nurses and nursing assistants. **Methods:** A prospective quasi-experimental study was conducted between 2022 and 2024 in three private hospitals in Madrid. A total of 200 newly hired health professionals (128 nurses and 72 assistants) were assigned alternately to either the intervention group (structured onboarding program) or the control group (usual routine). The intervention comprised three consecutive days of guided training with mentorship, simulation-based learning, and digital tool instruction. Adaptation was assessed with the validated GAML scale, and satisfaction was measured through a Likert survey one month later. Statistical analyses included Mann–Whitney U, Chi-squared tests, and linear regression. **Results:** The intervention group achieved significantly higher scores across all competency domains for both nurses and nursing assistants, with overall medians of 25 [22–27] and 22 [20–23.25], respectively, compared with notably lower values in the control groups (*p* < 0.001). The greatest improvements were observed in digital tool management, clinical protocol knowledge, problem-solving and decision-making, and patient safety practices, demonstrating the strong impact of the structured onboarding program. In terms of satisfaction, participants in the intervention group also reported higher ratings for the clarity and completeness of information, particularly regarding hospital structure, service-specific orientation, and occupational risk prevention. However, global satisfaction differences were more pronounced among nurses than nursing assistants. **Conclusions:** The structured onboarding program demonstrated substantial benefits in professional adaptation, safety culture, and perceived preparedness of newly hired staff. These findings support integrating standardized onboarding plans as part of hospital quality and safety strategies, requiring sustained leadership and resource investment for long-term success.

## 1. Introduction

The nursing profession is currently facing a complex situation marked by high staff turnover, a shortage of qualified personnel, and difficulties in adapting to units with specific characteristics [1]. Compounding these factors is the increasing physical and emotional exhaustion among staff, which affects both team stability and the quality and safety of care [2,3]. This reality poses a strategic challenge for healthcare institutions, which must ensure continuity of care while attracting and retaining talent amid intense pressure on healthcare services [4].

In this context, hospital onboarding plans are an essential tool [5]. These programs aim to facilitate the integration of newly hired professionals by offering a structured process that allows them to become familiar with the institutional culture, clinical protocols, and team dynamics [6]. Beyond being an administrative formality, onboarding plans are organizational strategies designed to reduce the learning curve, improve cohesion, and ensure the provision of quality and safe care [7,8].

The scientific literature shows that implementing onboarding plans has multiple benefits. Among the most relevant are improved clinical safety, increased confidence in professional performance, greater job satisfaction, reduced turnover, and consolidation of the organizational structure. These programs also help clarify roles, improve communication between teams, and promote a collaborative work environment—factors directly related to staff retention and quality of care [9,10,11].

The implementation of structured onboarding programs has been shown to reduce clinical errors, improve multidisciplinary cohesion, and optimize patient experience. These demonstrate that onboarding not only has an impact on the well-being of workers but also constitutes a strategic element for the sustainability of healthcare systems and the improvement in healthcare outcomes [12].

In Spain, onboarding programs have been developed in a heterogeneous manner. In many hospitals, they are limited to administrative procedures or a brief orientation to the assigned unit [13,14], while in other European countries or in the United States, more standardized programs are implemented that include mentoring, clinical simulation, and continuous assessment [12].

Although a few isolated studies have focused on the implementation of onboarding plans in our context, such as in a public hospital [15] or in a pediatric ICU, where the effectiveness of a theoretical–practical plan was evaluated [16], these are preliminary studies with a low number of participants.

This difference highlights the importance of implementing solid induction plans, as key factors such as patient safety, the smooth integration of professionals, and talent retention—particularly in a healthcare landscape marked by nursing shortages—depend on them. Therefore, there is a clear need for innovative proposals that expand on the traditional approach. The present study differs in this respect: it proposes a structured, measurable, intensive plan accompanied by mentoring, which incorporates key cross-cutting tools and skills for daily nursing work.

## 2. Materials and Methods

### 2.1. Aim

The aim of this study is to design and analyze the impact of an accelerated, structured onboarding program for newly hired healthcare staff. The main objective is to analyze whether an onboarding program, compared to the usual routine, influences early professional adaptation, safety culture, and the optimization of human resource management, contributing to the development of more efficient and sustainable organizational strategies. In addition, as a secondary objective, this study aims to analyze differences in the satisfaction among newly recruited nurses.

### 2.2. Study Design

This prospective quasi-experimental study, conducted between June 2022 and December 2024, employed a two-arm design (intervention and control) and a single-blinded approach (data analysts), comparing two conditions: the intervention group (IG), which received an onboarding program intervention, and the control group (CG), which underwent standard training.

This study involved 200 newly hired health professionals: 128 nurses and 72 nursing assistants. The newly hired participants included recent graduates and individuals with previous work experience. Both the nurse and nursing assistant groups comprised recent graduates, with the former including nurses with more than 10 years of experience, while the latter included those with up to 10 years of experience. All participants lacked prior experience with the internal protocols of the institution.

Participants were assigned to one of two groups using an alternating systematic method: the first participant was assigned to the CG (receiving standard training), while the second was assigned to the IG, and this pattern continued sequentially.

### 2.3. Study Setting and Sampling

This study was conducted across three private hospitals belonging to the HM Hospitales group: HM Montepríncipe, HM Torrelodones, and HM Sanchinarro. All three centers are located in Madrid and have the same nurse-to-patient ratio, admit patients with comparable internal medicine and surgical pathologies, and operate under standardized clinical protocols and procedures.

The inclusion and exclusion criteria are as follows:

Inclusion criteria:Participants were newly hired healthcare professionals with a valid nursing or auxiliary nursing degree.Participants had no prior working experience within any hospital of the HM Hospitales group.Participants had full-time contracts.

Exclusion criteria:Participants had part-time contracts.Participants had previously worked in the HM hospital group.

### 2.4. Intervention

The IG participants underwent the onboarding program guided by a nurse educator, delivered over three consecutive days with seven hours of structured training per day. Following recruitment, each new staff member was individually assigned a nurse educator. This mentor oversaw training for the following three days, ensuring continuity and personalized follow-up.

The program’s innovative approach lies in the combination of daily mentoring by a single trained nurse educator, the use of institutional guides and instructional videos, and the incorporation of defined learning objectives to facilitate progressive adaptation according to the varying competencies of the nurses and nurse assistants. This structured program offers a pedagogically designed experience supported by standardized resources, simulation, and deliberate assessment criteria.

Over three consecutive days, the onboarding program focused on institutional orientation, developing administrative and technical skills according to the institution’s digital platforms, and clinical integration (see Figure 1).

The CG received the usual onboarding approach consisting of three days of informal shadowing under an assigned nurse, followed by independent clinical practice.

### 2.5. Outcome Measures

#### 2.5.1. Adaptation to the Work Environment

In order to evaluate the impact of the intervention and the adaptation of the newly hired professionals to the hospital setting, an ad hoc tool was designed. Validated scales such as the Casey-Fink Surveys [17] and the Nursing Work Index [18] were reviewed. However, they did not fully apply to the context of the study, as they were more oriented toward the stress of transitioning from student to practice or the satisfaction and productivity of nurses than toward understanding whether they had acquired knowledge and skills in managing aspects intrinsic to the hospital. The Organizational Socialization Inventory (OSI) [19] was also reviewed, and its four components (training, understanding, peer support, and future perceptions) were analyzed. In this case, some aspects of this scale did not align with the study’s objectives and failed to capture information related to the intervention. Following this analysis, a new tool fully tailored to the evaluation of the intervention was designed.

Adaptation to the work environment among new workers was measured using the GAML instrument (Adaptation to the Work Environment Scale). This tool was designed by the researchers and validated through a consensus by the Nursing Management and the Quality Department of the Hospital. The instrument consists of 29 dichotomous items distributed across seven domains, as follows:The management of digital tools is a domain that evaluates the use of the hospital’s intranet and digital platforms to locate key documents such as nursing care protocols, occupational risk prevention documentation, and incident notification reporting through 6 questions.Knowledge of clinical protocols assesses the use of the institution’s digital platform for nursing care and reporting medical records through 5 questions.Compliance with administrative regulations focuses on the correct use of the platform for prescription, interpretation, and validation of pharmacological treatments for carrying out pharmaceutical reports through 4 questions. This domain is answered only by nurses.Decision-making and problem-solving is a domain that assesses knowledge of emergency protocols, emergency contact numbers, cardiopulmonary resuscitation (CPR) procedures, and evacuation plan reporting through 4 questions.Diet measures the ability to introduce dietary plans in the hospital system according to medical orders through 2 questions. This domain is answered only by nursing assistants.The domain patient safety practices assess the knowledge of patient safety, allergies, hygiene, and warning reports through 6 questions.Administrative skills is a domain that covers the use of expense forms and the correct entry of billing data through 2 questions.

Each item was scored as “pass” or “fail” (1/0). Participants were considered to have successfully completed the onboarding process if they met at least 80% of the required criteria—19 items for nurses, 18 for assistant nurses. Internal reliability was assessed using the KR-20 coefficient, yielding a score of 0.91, indicating excellent consistency.

#### 2.5.2. Sociodemographic Data

Age, sex, nationality, and previous experience were collected as sociodemographic data.

#### 2.5.3. Satisfaction

Additionally, a satisfaction survey was administered one month after onboarding using a 5-point Likert scale to assess the clarity of the process and the quality of mentorship through 8 questions.

### 2.6. Data Collection

The study protocol followed a structured process: once the Human Resources department notified the Nursing Director of the hiring of a new nurse or nursing assistant, the latter would assign the new hire to the necessary hospital unit based on their experience. In addition, using an alternating systematic method, the Director of Nursing would identify the participant as either a CG, who would receive standard training, or an IG, who would undertake the structured onboarding program.

Afterwards, the Director of Nursing would inform the Unit Supervisor of the new hire and, in the case of the IG, the nursing trainer would be informed to follow the three-day instructions. In the IG, the supervisor and trainer would welcome the new hire. The trainer guides and trains them during the three days of intervention. Then, all participants (from the IG and the CG) must complete the GAML questionnaire and fill in their sociodemographic data on paper. One month following this, a satisfaction survey is sent via corporate email. New professionals are informed that their hiring did not depend on these evaluations and that their sole purpose is to identify areas for improvement in the onboarding process.

### 2.7. Data Analysis

All data were analyzed using SPSS version 23. The significance of the data was established based on a confidence level of 95% (*p* < 0.05). Quantitative variables were described using medians and interquartile ranges (IQRs), while categorical variables were expressed in absolute frequencies and percentages. The Mann–Whitney U test was applied to compare the total GAML scores between groups and the satisfaction test. Chi-squared tests were used to test our hypotheses for the qualitative variables.

### 2.8. Ethical Considerations

The study was approved by the HM Hospitales Research Ethics Committee (Ref. No. 21.06.1860–GHM).

All data were handled anonymously and confidentially in accordance with European Regulation (EU) 2016/679 on data protection, Organic Law 3/2018 on personal data and digital rights, and Law 14/2007 on biomedical research. Only the principal investigator had access to the encrypted database.

## 3. Results

The sample of 128 nurses was distributed equally between the CG (n = 64) and the IG (n = 64). The median age was similar in both groups, with no statistically significant differences (25 years [24–27] in the CG vs. 24 [23–28.25] in the UG; *p* = 0.061). Almost all participants were women (99.2%), with no differences between groups (*p* = 1.000). In terms of nationality, most were of Spanish origin (78.9%), followed by Chilean (11.7%), with no significant differences between groups (*p* = 0.530). Previous work experience showed a median of 12 months in the total sample, with similar values between groups (12.5 months [9.75–18.00] in the control and 9.0 [2.00–25.25] in the intervention; *p* = 0.196). Overall, no statistically significant differences were observed in the sociodemographic variables between the control and IGs of nurses (see Table 1).

Table 2 presents data on 72 nursing assistants, equally distributed between the CG (n = 36) and the IG (n = 36). Statistically significant differences were observed in age, with participants in the IG being older (25 [24–26] years) than those in the CG (20 [18–21] years; *p* < 0.001). Most were women (84.7%), with no significant differences between groups (*p* = 1.000). In terms of nationality, Spanish origin predominated (79.2%), although differences were observed between groups (*p* = 0.018), with a greater presence of people from Colombia and Venezuela in the IG. Previous work experience was also significantly higher in the IG (6 [6–15] months) than in the CG (1.5 [0–9] months; *p* = 0.001).

As shown in Table 3, the comparative analysis of competencies among nurses measured via the GAML tool demonstrates a consistent and statistically significant advantage for the IG, which received the specific training program, across nearly all evaluated domains. Globally, the IG achieved a markedly higher median score (25 [22–27]) compared with the CG (12.5 [10–15]; *p* < 0.001), underscoring the overall effectiveness of the onboarding plan.

The most notable improvements were observed in the management of digital tools, where the IG attained a median of 5 [4–6] versus 1 [0–3] in CG (*p* < 0.001). Subdomains such as nursing care documentation, procedures, protocols, and incident reporting showed particularly robust differences, with absolute increments exceeding 50 percentage points in favor of the trained group. This finding reflects not only improved familiarity with digital platforms but also enhanced application to clinical practice.

Similarly, knowledge of clinical protocols was significantly superior in the IG, with median scores at the maximum value of 5, compared with 4 [2.75–5] in the control (*p* < 0.001). All assessed competencies as vital sign registration, pain scales, and management of laboratory and radiological tests, reached near-universal proficiency in the IG (≥89%), in contrast with lower rates in the CG (59–73%).

Competencies in problem-solving and decision-making also showed large differences (median 4 vs. 0; *p* < 0.001), particularly in emergency-related items such as CPR protocols and evacuation procedures, where trained participants more than doubled the performance rates of controls. Furthermore, patient safety was substantially reinforced by the training program, with significant improvements in the identification of patients, allergy documentation, and adherence to safety guidelines. By contrast, domains such as adherence to administrative regulations and billing also showed significant but comparatively smaller improvements, while areas with high baseline performance (e.g., treatment validation) demonstrated ceiling effects.

As shown in Table 4, a similar pattern was observed among nursing assistants, with the IG outperforming the CG across almost all GAML domains. The global competency score was significantly higher in the IG (22 [20–23.25]) than in the CG (15.5 [12–17.25]; *p* < 0.001), confirming the program’s impact across different professional categories.

In the management of the digital tools domain, the IG exhibited a median of 5 [5–6] compared to 3 [1–4] in the control (*p* < 0.001). Large gaps were observed in the correct use of protocols, documentation of procedures, and incident management, indicating that the program effectively strengthened digital literacy and procedural accuracy.

Knowledge of clinical protocols followed the same trend (median 5 [4–5] vs. 4 [2–4]; *p* < 0.001), with trained assistants demonstrating near-perfect adherence in vital sign and pain assessment documentation, and in the management of laboratory and imaging tests.

Improvements were also pronounced in problem-solving and decision-making (median 3 [2–3] vs. 1 [0–2]; *p* < 0.001), especially for emergency management and evacuation protocols, highlighting greater preparedness and confidence in critical situations.

The patient safety domain likewise showed significant advances (median 6 [5–6] vs. 4 [3–5]; *p* < 0.001). The IG demonstrated better compliance with the “10 rights” of patient care, proper use of digital platforms, and delivery of safety-related advice to patients, all key indicators of safe clinical practice.

Conversely, domains such as diet management and billing did not differ significantly between groups, likely due to uniformly high baseline performance or limited scope for improvement in these specific tasks.

Of the 200 participants, 132 responded to the satisfaction questionnaire one month later. The analysis revealed that the IG reported higher satisfaction scores compared with the CG, particularly among nurses.

As shown in Table 5, nurses in the IG reported significantly higher satisfaction levels than those in the CG across most domains. The IG showed better ratings in the quality and amount of information received, especially regarding general hospital information, structure, service-specific details, pharmacy service, occupational risk prevention, and work unit information (all *p* < 0.001). These results reflect that the specific onboarding program effectively provided clearer, more comprehensive, and better-structured information, facilitating integration and confidence among newly hired nurses. Overall satisfaction was also higher in the IG (median 4 [3–5]) compared with the CG (3 [3–4]; *p* = 0.015), confirming the positive impact of the training plan on the adaptation process.

According to Table 6, nursing assistants in the IG also reported greater satisfaction in several aspects of the onboarding experience, particularly in information about the hospital structure (*p* = 0.029), pharmacy service (*p* < 0.001), occupational risk prevention (*p* < 0.001), and work unit information (*p* = 0.018). These improvements indicate that the specific plan enhanced communication and information accessibility, especially in safety and operational domains. However, overall satisfaction did not differ significantly between groups (*p* = 0.552).

## 4. Discussion

Our study provides evidence supporting the value of structured induction programs in high-complexity hospitals in Spain with positive results. Studies published to date in our context differ in terms of hospital type (public vs. private) and size (hospital vs. unit) from ours. Most published studies implement measures to onboard mainly newly graduated nurses, while our study is aimed at newly hired nurses, whether they are recent graduates or not [10,20]. This highlights the importance of designing training structures that improve professionals’ adaptation to a new job in a hospital setting.

Our findings are in line with previous research linking such interventions to improved staff integration, learning, knowledge, adherence to clinical protocols, communication, built teamwork, self-confidence in patient safety, and satisfaction [21].

The IG obtained significantly higher scores in almost all domains evaluated by the GAML tool compared to the CG. These include digital tool management, care documentation, clinical protocols, decision-making, and patient safety. This suggests that specific training improves both theoretical knowledge and practical skills and accelerates adaptation to the hospital environment.

Some studies highlighted how structured hospital processes can reduce clinical errors and facilitate staff adaptation. Our findings reinforce this notion: the IG of both nurses and nurse assistants demonstrated superior clinical performance, particularly in patient safety and emergency management domains. These results may be related to the incorporation of clinical simulations and mentoring strategies [22,23].

In addition, digital integration appears to play a substantial role in effective onboarding. Research shows that the use of structured e-learning platforms and digital simulation tools enhances comprehension of hospital workflows and safety practices [24]. Our structured program, which includes digital modules and scenario-based evaluations, may thus reflect broader trends in modern clinical education.

The workplace environment also proved to be a critical factor for successful onboarding. According to Requeno et al. [25], early emotional support and clear expectations foster a greater sense of belonging among healthcare workers. In our study, a senior nurse accompanied the newly hired nurses during their training, which helped them integrate into the team. The basis for the successful integration of new hospital workers involves developing a good induction plan based on integration with the team and a sense of belonging [10].

From an organizational perspective, one of the most relevant findings was the high satisfaction with the program one month later. This aligns with previous studies indicating that structured mentorship and early support programs contribute to improved job satisfaction and long-term retention [26,27]. Specifically, consistent feedback, team engagement, and early exposure to institutional culture are pivotal elements in reducing professional disaffection [28].

Lastly, implementing such structured programs requires institutional commitment and resource allocation. Effective onboarding models, while impactful, depend on ongoing investment and leadership support. Scalability and adaptability remain key challenges for widespread adoption across different healthcare settings [29,30].

Despite these positive findings, this study has several limitations. The non-random assignment of participants may introduce selection bias. Additionally, the quasi-experimental design does not allow for definitive causal inferences. We use The Transparent Reporting of Evaluations with Nonrandomized Designs (TREND) declaration checklist [31] for nonrandomized research. Finally, contamination between groups cannot be ruled out, and external validation of the GAML tool remains pending, despite strong internal consistency (KR-20 = 0.91).

## 5. Conclusions

The onboarding plan with specific training evaluated in this study demonstrated clear and strong effects on multiple clinical, administrative, and safety competencies, as well as on the satisfaction of new nurses and nurse assistants. For these benefits to be sustained and widely adopted, sustained investment, institutional leadership commitment, and adaptation to the local context are required. The results support that structured onboarding programs should be part of quality and safety strategies in hospitals, rather than being confined solely to human resources policy.

## Figures and Tables

**Figure 1 nursrep-15-00398-f001:**
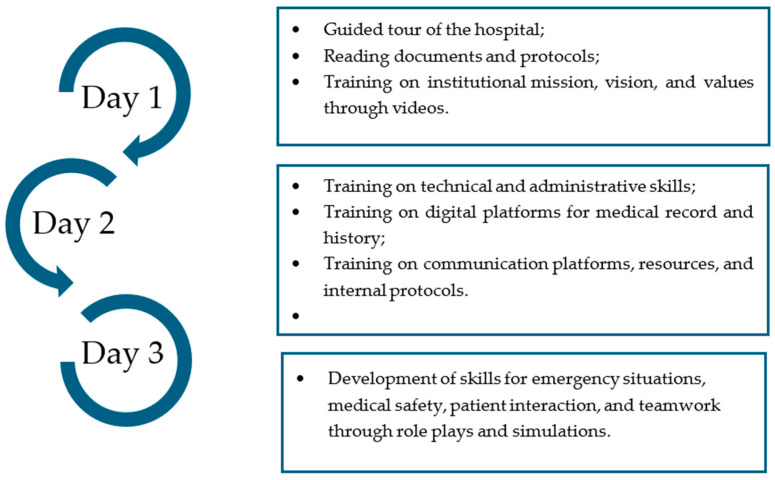
Structured onboarding program.

**Table 1 nursrep-15-00398-t001:** Sociodemographic data of nurses.

		Global (n = 128)	Control (n = 64)	Intervention (n = 64)	*p*-Value
Age (median, IQR)		25.00 [23.00;28.00]	25.00 [24.00;27.00]	24.00 [23.00;28.25]	0.061
Sex (n, %)	Women	127 (99.22%)	64 (100.00%)	63 (98.44%)	1.000
Nationality (n, %)	Chile	15 (11.72%)	6 (9.38%)	9 (14.06%)	0.530
	Colombia	4 (3.12%)	2 (3.12%)	2 (3.12%)	
	Spain	101 (78.91%)	52 (81.25%)	49 (76.56%)	
	Peru	4 (3.12%)	3 (4.69%)	1 (1.56%)	
	Romania	2 (1.56%)	0 (0.00%)	2 (3.12%)	
	Ukraine	1 (0.78%)	1 (1.56%)	0 (0.00%)	
	Venezuela	1 (0.78%)	0 (0.00%)	1 (1.56%)	
Previous Experience (months; median, IQR)		12.00 [5.75;24.00]	12.50 [9.75;18.00]	9.00 [2.00;25.25]	0.196

IQR: Interquartile range.

**Table 2 nursrep-15-00398-t002:** Sociodemographic data of nurse assistants.

		Global (n = 72)	Control (n = 36)	Intervention (n = 36)	*p*-Value
Age (median, IQR)		23.00 [19.75;25.00]	20.00 [18.00;21.00]	25.00 [24.00;26.00]	<0.001
Sex (n, %)	Women	61 (84.72%)	31 (86.11%)	30 (83.33%)	1.000
Nationality (n, %)	Chile	6 (8.33%)	4 (11.11%)	2 (5.56%)	0.018
	Colombia	4 (5.56%)	0 (0.00%)	4 (11.11%)	
	Spain	57 (79.17%)	31 (86.11%)	26 (72.22%)	
	Romania	1 (1.39%)	1 (2.78%)	0 (0.00%)	
	Venezuela	4 (5.56%)	0 (0.00%)	4 (11.11%)	
Previous Experience (months; median, IQR)		6.00 [0.75;12.00]	1.50 [0.00;9.00]	6.00 [6.00;15.00]	0.001

IQR: Interquartile rank.

**Table 3 nursrep-15-00398-t003:** Difference between groups by training program and competency of nurses.

Domains of the GAML Tool	Total (n = 128)	Control (n = 64)	Intervention (n = 64)	*p*-Value
Global (median, IQR)	19.00 [12.75;25.00]	12.50 [10.00;15.00]	25.00 [22.00;27.00]	<0.001
Management of digital tools (median, IQR)	3.00 [1.00;5.00]	1.00 [0.00;3.00]	5.00 [4.00;6.00]	<0.001
Informed consent (n, %)	77 (60.16%)	17 (26.56%)	60 (93.75%)	<0.001
Nursing care (n, %)	77 (60.16%)	21 (32.81%)	56 (87.50%)	<0.001
Procedures (n, %)	77 (60.16%)	21 (32.81%)	56 (87.50%)	<0.001
Protocols (n, %)	55 (42.97%)	7 (10.94%)	48 (75.00%)	<0.001
Occupational risk prevention (n, %)	38 (29.69%)	0 (0.00%)	38 (59.38%)	<0.001
Incidents (n, %)	70 (54.69%)	19 (29.69%)	51 (79.69%)	<0.001
Knowledge of clinical protocols (median, IQR)	5.00 [4.00;5.00]	4.00 [3.00;5.00]	5.00 [5.00;5.00]	<0.001
Vital sign register (n, %)	104 (81.25%)	40 (62.50%)	64 (100.00%)	<0.001
Pain register (n, %)	105 (82.03%)	41 (64.06%)	64 (100.00%)	<0.001
Evolutive register (n, %)	123 (96.09%)	59 (92.19%)	64 (100.00%)	0.058
Lab test (n, %)	111 (86.72%)	47 (73.44%)	64 (100.00%)	<0.001
Radiology test management (n, %)	103 (80.47%)	47 (73.44%)	56 (87.50%)	0.074
Adherence to administrative regulations (median, IQR)	4.00 [4.00;4.00]	3.00 [3.00;4.00]	4.00 [4.00;4.00]	0.001
Medication guidelines (n, %)	118 (92.19%)	55 (85.94%)	63 (98.44%)	0.021
Pharmacy incidence (allergies) (n, %)	117 (91.41%)	54 (84.38%)	63 (98.44%)	0.012
Treatment interpretation (n, %)	122 (95.31%)	58 (90.62%)	64 (100.00%)	0.028
Treatment validation (n, %)	125 (97.66%)	61 (95.31%)	64 (100.00%)	0.244
Problem-solving and decision-making (median, IQR)	1.00 [0.00;4.00]	0.00 [0.00;1.00]	4.00 [3.00;4.00]	<0.001
CPR protocol (n, %)	76 (59.38%)	22 (34.38%)	54 (84.38%)	<0.001
CPR circuit (n, %)	60 (46.88%)	4 (6.25%)	56 (87.50%)	<0.001
Emergency advice (n, %)	64 (50.00%)	8 (12.50%)	56 (87.50%)	<0.001
Evacuation protocol (n, %)	35 (27.34%)	0 (0.00%)	35 (54.69%)	<0.001
Patient safety (median, IQR)	5.00 [4.00;6.00]	4.00 [2.00;4.00]	6.00 [5.00;6.00]	<0.001
Platform skills (n, %)	52 (40.62%)	3 (4.69%)	49 (76.56%)	<0.001
10 rights (n, %)	107 (83.59%)	45 (70.31%)	62 (96.88%)	<0.001
Allergies identification (n, %)	95 (74.22%)	31 (48.44%)	64 (100.00%)	<0.001
Hand hygiene (n, %)	116 (90.62%)	54 (84.38%)	62 (96.88%)	0.034
Advice (n, %)	99 (77.34%)	35 (54.69%)	64 (100.00%)	<0.001
Patient identification (n, %)	85 (66.41%)	22 (34.38%)	63 (98.44%)	<0.001
Billing (median, IQR)	1.00 [0.00;2.00]	0.00 [0.00;0.00]	2.00 [1.00;2.00]	<0.001
Expense report (n, %)	63 (49.22%)	9 (14.06%)	54 (84.38%)	<0.001
Billing control (n, %)	53 (41.41%)	5 (7.81%)	48 (75.00%)	<0.001

GAML: Adaptation to the Work Environment Scale; IQR: Interquartile rank.

**Table 4 nursrep-15-00398-t004:** Difference between groups by training program and competency of nursing assistants.

Domains of the GAML Tool	Total (n = 72)	Control (n = 36)	Intervention (n = 36)	*p*-Value
Global (median, IQR)	19.00 [15.00;22.00]	15.50 [12.00;17.25]	22.00 [20.00;23.25]	<0.001
Management of digital tools (median, IQR)	4.00 [3.00;5.00]	3.00 [1.00;4.00]	5.00 [5.00;6.00]	<0.001
Informed consent (n, %)	51 (70.83%)	17 (47.22%)	34 (94.44%)	<0.001
Nursing care (n, %)	45 (62.50%)	15 (41.67%)	30 (83.33%)	<0.001
Procedures (n, %)	53 (73.61%)	19 (52.78%)	34 (94.44%)	<0.001
Protocols (n, %)	45 (62.50%)	14 (38.89%)	31 (86.11%)	<0.001
Occupational risk prevention (n, %)	26 (36.11%)	8 (22.22%)	18 (50.00%)	0.027
Incidents (n, %)	55 (76.39%)	22 (61.11%)	33 (91.67%)	0.006
Knowledge of clinical protocols (median, IQR)	4.00 [3.75;5.00]	4.00 [2.00;4.00]	5.00 [4.00;5.00]	<0.001
Vital sign register (n, %)	55 (76.39%)	19 (52.78%)	36 (100.00%)	<0.001
Pain register (n, %)	59 (81.94%)	24 (66.67%)	35 (97.22%)	0.002
Evolutive register (n, %)	39 (54.17%)	14 (38.89%)	25 (69.44%)	0.018
Lab test (n, %)	60 (83.33%)	25 (69.44%)	35 (97.22%)	0.004
Radiology test management (n, %)	57 (79.17%)	21 (58.33%)	36 (100.00%)	<0.001
Problem-solving and decision-making (median, IQR)	2.00 [1.00;3.00]	1.00 [0.00;2.00]	3.00 [2.00;3.00]	<0.001
CPR protocol (n, %)	50 (69.44%)	16 (44.44%)	34 (94.44%)	<0.001
CPR circuit (n, %)	2 (2.78%)	0 (0.00%)	2 (5.56%)	0.493
Emergency advice (n, %)	53 (73.61%)	19 (52.78%)	34 (94.44%)	<0.001
Evacuation protocol (n, %)	28 (38.89%)	6 (16.67%)	22 (61.11%)	<0.001
Diet (median, IQR)	2.00 [2.00;2.00]	2.00 [2.00;2.00]	2.00 [2.00;2.00]	0.160
Allergy’s introduction (n, %)	70 (97.22%)	34 (94.44%)	36 (100.00%)	0.493
program management (n, %)	70 (97.22%)	34 (94.44%)	36 (100.00%)	0.493
Patient safety (median, IQR)	5.00 [4.00;6.00]	4.00 [3.00;5.00]	6.00 [5.00;6.00]	<0.001
Platform skills (n, %)	33 (45.83%)	7 (19.44%)	26 (72.22%)	<0.001
10 rights (n, %)	46 (63.89%)	13 (36.11%)	33 (91.67%)	<0.001
Allergies identification (n, %)	65 (90.28%)	30 (83.33%)	35 (97.22%)	0.107
Hand hygiene (n, %)	68 (94.44%)	32 (88.89%)	36 (100.00%)	0.115
Advice (n, %)	60 (83.33%)	26 (72.22%)	34 (94.44%)	0.027
Patient identification (n, %)	65 (90.28%)	30 (83.33%)	35 (97.22%)	0.107
Billing (median, IQR)	2.00 [1.00;2.00]	2.00 [2.00;2.00]	2.00 [1.00;2.00]	0.093
Expense report (n, %)	71 (98.61%)	35 (97.22%)	36 (100.00%)	1
Billing control (n, %)	58 (80.56%)	32 (88.89%)	26 (72.22%)	0.137

GAML: Adaptation to the Work Environment Scale; IQR: Interquartile rank.

**Table 5 nursrep-15-00398-t005:** Difference between groups according to nurse satisfaction.

Satisfaction Variables	Total (n = 92)	Control (n = 44)	Intervention (n = 48)	*p*-Value
Information quality				
General information received about the hospital (median, IQR)	4.00 [4.00;5.00]	4.00 [3.00;4.00]	5.00 [4.00;5.00]	<0.001
Information about the structure of the hospital (median, IQR)	4.00 [3.00;5.00]	3.00 [3.00;4.00]	4.00 [4.00;5.00]	<0.001
Information about the work (median, IQR)	4.00 [4.00;5.00]	4.00 [4.00;5.00]	4.00 [4.00;5.00]	0.087
Service-specific information (median, IQR)	4.00 [3.00;4.00]	3.00 [3.00;4.00]	4.00 [4.00;5.00]	<0.001
Amount information				
Information about digital platform (median, IQR)	4.00 [4.00;5.00]	4.00 [3.00;5.00]	4.00 [4.00;5.00]	0.057
Information about the structure of the hospital (median, IQR)	4.00 [3.00;5.00]	3.00 [3.00;4.00]	4.00 [3.75;5.00]	0.012
Information about the pharmacy service (median, IQR)	3.00 [2.00;4.00]	2.00 [1.00;2.00]	4.00 [3.00;5.00]	<0.001
Information about occupational risk prevention (median, IQR)	2.00 [0.00;4.00]	0.00 [0.00;0.00]	4.00 [3.00;5.00]	<0.001
Information about work unit (median, IQR)	4.00 [3.00;5.00]	3.00 [3.00;4.00]	5.00 [4.00;5.00]	<0.001
Total satisfaction (median, IQR)	4.00 [3.00;5.00]	3.00 [3.00;4.00]	4.00 [3.00;5.00]	0.015

IQR: Interquartile range.

**Table 6 nursrep-15-00398-t006:** Difference between groups according to nursing assistants’ satisfaction.

Satisfaction Variables	Total (n = 45)	Control (n = 31)	Intervention (n = 14)	*p*-Value
Information quality				
General information received about the hospital (median, IQR)	4.00 [4.00;5.00]	4.00 [3.00;5.00]	4.50 [4.00;5.00]	0.134
Information about the structure of the hospital (median, IQR)	4.00 [3.00;4.00]	4.00 [3.00;4.00]	4.00 [4.00;5.00]	0.029
Information about the work (median, IQR)	4.00 [4.00;5.00]	4.00 [4.00;5.00]	5.00 [4.00;5.00]	0.337
Service-specific information (median, IQR)	4.00 [3.00;5.00]	4.00 [3.00;4.00]	4.00 [4.00;5.00]	0.139
Amount information				
Information about digital platform (median, IQR)	5.00 [4.00;5.00]	5.00 [4.00;5.00]	5.00 [4.00;5.00]	0.826
Information about the structure of the hospital (median, IQR)	4.00 [3.00;5.00]	4.00 [3.00;5.00]	4.00 [4.00;5.00]	0.170
Information about the pharmacy service (median, IQR)	2.00 [2.00;3.00]	2.00 [1.00;2.00]	4.00 [3.00;5.00]	<0.001
Information about occupational risk prevention (median, IQR)	0.00 [0.00;3.00]	0.00 [0.00;0.00]	4.00 [3.00;5.00]	<0.001
Information about work unit (median, IQR)	4.00 [3.00;5.00]	3.00 [3.00;4.00]	5.00 [4.00;5.00]	0.018
Total satisfaction (median, IQR)	4.00 [3.00;5.00]	5.00 [3.50;5.00]	4.00 [3.00;5.00]	0.552

IQR: Interquartile range.

## Data Availability

The data are available through the corresponding author.

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
