# Peer review of "The Impact of an Onboarding Plan for Newly Hired Nurses and Nursing Assistants: Results of a Quasi-Experimental Study"

_nursrep, 2025, doi:10.3390/nursrep15110398_

Round 1

Reviewer 1 Report

Comments and Suggestions for Authors

Dear Authors,
Congratulations on addressing this topic; the professional adaptation of new employees is a very important issue. Its proper course determines not only job satisfaction but also guarantees lower absenteeism due to illness and significantly lower employee turnover related to leaving the profession and professional burnout.
However, I suggest introducing a few changes to the document text.
1. Comparing the professional adaptation of nursing assistants and nurses with varying levels of professional experience may lead to misinterpretation. The requirements for these groups are different, and the adaptation process itself must also differ. I suggest considering conducting separate analyses. 
2. Several questions also arise in connection with the analysis of the collected data:
- Does the adaptation time differ or is it the same depending on the level of experience and the nature of the work of a nursing assistant and a nurse?
- What is the structure of these groups, which by definition cannot be treated as homogeneous (this suggests describing the characteristics of the groups separately in a table).
 - Similarly, differences between nurses participating in the program and those not participating in the program should be presented separately. Presenting data for all medical employees in one table,  is not necessary and does not contribute to the better  interpretation of the relationship, and it may even make it more difficult and take misstake interpretion  (Table 2), similarly in Table 3.
 - When presenting a general multiple regression model, the model components should be described and more precisely presented in a table, preceded by a verification of the interactions between the tested model components. Was this performed? ,There is no information in the document.
 - When presenting abbreviations in the paper, they should be fully described upon first use; this was not presented clearly enough in the paper.  The discussion should refer to the results of the paper.
Best regards, Reviewer

Author Response

Please see the attached document with responses to the reviewer's comments.

Reviewers  Comments

Answers

1. Comparing the professional adaptation of nursing assistants and nurses with varying levels of professional experience may lead to misinterpretation. The requirements for these groups are different, and the adaptation process itself must also differ. I suggest considering conducting separate analyses.

Thank you very much for your comment We completely agree with you. Indeed, the characteristics are different, as is the adaptation process. Therefore, in the results, we have separated the results for nurses and nursing assistants. This makes the impact of the induction plan on both groups clearer. You can find the change in the results section.

Several questions also arise in connection with the analysis of the collected data:
- Does the adaptation time differ or is it the same depending on the level of experience and the nature of the work of a nursing assistant and a nurse?

The adaptation time was the same for nurses and assistants, lasting three days. However, although the methodology was the same, the content was different since the two professionals' jobs differ. We had better explain it in line 132.

 What is the structure of these groups, which by definition cannot be treated as homogeneous (this suggests describing the characteristics of the groups separately in a table).

According to your comment we have reported the data separately in two tables (Table 1 and Table 2).

 Similarly, differences between nurses participating in the program and those not participating in the program should be presented separately. Presenting data for all medical employees in one table,  is not necessary and does not contribute to the better  interpretation of the relationship, and it may even make it more difficult and take misstake interpretion  (Table 2), similarly in Table 3.

Thank you for your feedback. We have separated the questionnaire results report into Table 3 and Table 4, as well as the satisfaction report into Table 5 and Table 6. This allows us to delve deeper into the impact of the results by professional type.

 When presenting a general multiple regression model, the model components should be described and more precisely presented in a table, preceded by a verification of the interactions between the tested model components. Was this performed? ,There is no information in the document.

After separating the analyses according to the nature of the professionals, linear regression models were again performed. However, after this analysis, the results were not clinically relevant, so their exposure was rejected.

When presenting abbreviations in the paper, they should be fully described upon first use; this was not presented clearly enough in the paper.  

Abbreviations have been revised and described at first use.

The discussion should refer to the results of the paper

We have increased the referred to our results in lines 336-348

Reviewer 2 Report

Comments and Suggestions for Authors
  • Lines 55–58

In the paragraph beginning with “The implementation of structured onboarding programs has been shown to reduce clinical errors…”, the placement of the citation is unclear. It seems that either an additional reference is missing at the end of the paragraph or that reference [12] should be moved so it supports the entire statement. Please review and correct this for clarity.

  • Line 86 (Study Design)

It would enhance transparency to specify the exact months or period during which the study was conducted (for example, January 2022 to March 2024). If possible, please clarify whether the participants were new graduates or had prior professional experience, and highlight any differences between nurses and nursing assistants, as these factors could have influenced adaptation outcomes.

  • Line 252

The sentence “This study is the first, as far as we know, that provides evidence supporting the value of structured onboarding programs in high-complexity hospitals in Spain” appears somewhat overstated. In your own reference list, items [15] and [16] (Ortega-Arias, 2024, Líderes Cuidados; and Enfermería Intensiva, 2022) describe similar onboarding initiatives, although conducted in different contexts—a public hospital and a pediatric ICU, respectively. It would be advisable to moderate this claim or rephrase it to acknowledge these existing studies, while clarifying what makes your work distinct or novel in relation to them.

  • Lines 169–177 (Data collection)

The description “The supervisor contacted the participant with the nurse educator to follow the three-day instructions…” could be clearer. It is not entirely evident who initiated the contact and when the GAML questionnaire and sociodemographic data were collected. A more precise explanation of this process would improve the reader’s understanding and the study’s reproducibility.

  • Lines 134–160 (GAML instrument)

The GAML instrument is described as “designed by the researchers and validated by consensus by the Nursing Management and the Quality Department.” While this internal review provides some degree of face validity, it is insufficient to establish full psychometric validation.

Although this limitation is briefly mentioned at the end of the Discussion, it should also be stated explicitly in the Methods section, with a short explanation of why an ad hoc instrument was created rather than adapting a pre-existing, validated tool.

There are several well-known instruments that assess adaptation and integration in nursing contexts, for instance:

  • Casey-Fink Graduate Nurse Experience Survey (Casey et al., 2004)
  • Nurse Work Index–Revised (NWI-R) (Aiken & Patrician, 2000)
  • Organizational Socialization Inventory (OSI) (Taormina, 1994)

Including a concise rationale for the development of GAML, and clarifying how it conceptually aligns or differs from these existing tools, would strengthen methodological transparency and enhance the overall rigor of the manuscript.

Comments on the Quality of English Language

English is generally understandable, the writing could be improved to achieve greater fluency and naturalness. A careful language review by a native English speaker or professional editor is recommended to ensure clarity and consistency throughout the text.

Author Response

Reviewer comments

Answer

·       Lines 55–58

In the paragraph beginning with “The implementation of structured onboarding programs has been shown to reduce clinical errors…”, the placement of the citation is unclear. It seems that either an additional reference is missing at the end of the paragraph or that reference [12] should be moved so it supports the entire statement. Please review and correct this for clarity.

Reference 12 has been moved to the end of the paragraph as it includes support for everything indicated above.

·       Line 86 (Study Design)

It would enhance transparency to specify the exact months or period during which the study was conducted (for example, January 2022 to March 2024). If possible, please clarify whether the participants were new graduates or had prior professional experience, and highlight any differences between nurses and nursing assistants, as these factors could have influenced adaptation outcomes

Thank you very much for your comments.

The exact data collection periods have been included. See line 94-95.

We have included information on the status of nurses and assistants with respect to their previous experience in this section. See lines 99 to 104.

We have also detailed this information in the results. Following your comment and the suggestion of the other reviewer, we have separated the results between registered nurses and nursing assistants. See results section.

·       Line 252

The sentence “This study is the first, as far as we know, that provides evidence supporting the value of structured onboarding programs in high-complexity hospitals in Spain” appears somewhat overstated. In your own reference list, items [15] and [16] (Ortega-Arias, 2024, Líderes Cuidados; and Enfermería Intensiva, 2022) describe similar onboarding initiatives, although conducted in different contexts—a public hospital and a pediatric ICU, respectively. It would be advisable to moderate this claim or rephrase it to acknowledge these existing studies, while clarifying what makes your work distinct or novel in relation to them.

We completely agree with this. Thank you very much for your comment. We have toned down the expression and reinforced our results in comparison with the contextual ones. See lines 337-344

  • Lines 169–177 (Data collection)

The description “The supervisor contacted the participant with the nurse educator to follow the three-day instructions…” could be clearer. It is not entirely evident who initiated the contact and when the GAML questionnaire and sociodemographic data were collected. A more precise explanation of this process would improve the reader’s understanding and the study’s reproducibility.

Thank you very much. We've included more detailed information about the process. See lines 197 to 211.

·       Lines 134–160 (GAML instrument)

The GAML instrument is described as “designed by the researchers and validated by consensus by the Nursing Management and the Quality Department.” While this internal review provides some degree of face validity, it is insufficient to establish full psychometric validation.

Although this limitation is briefly mentioned at the end of the Discussion, it should also be stated explicitly in the Methods section, with a short explanation of why an ad hoc instrument was created rather than adapting a pre-existing, validated tool.

There are several well-known instruments that assess adaptation and integration in nursing contexts, for instance:

·       Casey-Fink Graduate Nurse Experience Survey (Casey et al., 2004)

·       Nurse Work Index–Revised (NWI-R) (Aiken & Patrician, 2000)

·       Organizational Socialization Inventory (OSI) (Taormina, 1994)

Including a concise rationale for the development of GAML, and clarifying how it conceptually aligns or differs from these existing tools, would strengthen methodological transparency and enhance the overall rigor of the manuscript.

Thank you very much for your comment. We fully agree with you and have included these three references in the methodology section, alluding to the reason why they were not used. See line 148 to 159.

English is generally understandable, the writing could be improved to achieve greater fluency and naturalness. A careful language review by a native English speaker or professional editor is recommended to ensure clarity and consistency throughout the text.

Thank you for your feedback. We have sent the article to the journal's English review. Any unclear aspects have been corrected, making the text clearer and more coherent.
